# Evaluating LLM In-Context Few-Shot Learning on Legal Entity Annotation Task

## Abstract

The emergence of Large Language Models (LLMs) has attracted attention due to their powerful in-context few-shot learning capability. Recent studies present significant results regarding its usage in document annotation tasks; in some cases, the model is comparable to human annotators. In our work, we evaluate LLM's in-context few-shot learning capability on a legal NER, assessing its usage in an annotation task process with humans. To do so, our study is based on the most extensive corpus known in Portuguese dedicated to the legal NER. In our experiments, we tried six different LLMs in various setups, and the results showed that an LLM can produce highly accurate annotations; the best model achieved an **F1 score of 0.76**. Moreover, through a detailed manual inspection of the divergence cases, we identified opportunities for improvement in the annotation process that produced the corpus, with a significant portion of these issues being correctly addressed by the evaluated models. Thus, our results show that the models can assist annotators, reducing the time, effort, and errors produced during the annotation process.

## 1 Introduction

A considerable amount of legal documents is available on the Internet nowadays. Even so, knowledge extraction activities, such as Named Entity Recognition (NER), in the legal domain remain challenging, especially when dealing with non-English languages. One of the reasons is the lack of annotated corpora available, combined with the burden and cost of developing new ones. To address this gap in Portuguese, both Correia et al. (2022) and Brito et al. (2023) developed a corpus for legal NER, comprising decisions issued by the Brazilian Judiciary. Similar efforts include works such as Cao et al. (2022) and Leitner et al. (2020). Common to these initiatives are the presence of annotators and their subsequent costs, which can become unsustainable when dealing with high-volume or domain-specific data. In Correia et al. (2022), for instance, 95 law students were involved in the annotation task that lasted for months under close supervision, while in Brito et al. (2023), the initial annotation phase lasted two months and involved 36 legal experts, including public prosecutors and judges. Although none of these works mention the cost related to the annotation task, it is reasonable to assume that cost is a limiting factor.

Recent NLP advancements shifted from task-specific to task-agnostic approaches with Large Language Models (LLMs). Those models gained popularity due to their effectiveness in learning general representations over unlabeled data that can later be fine-tuned to perform a specific task (Brown et al., 2020) or even used in a few- or zero-shot way (Brown et al., 2020; Radford et al., 2019). In this sense, LLMs demonstrate powerful in-context few-shot learning capability on many text annotation tasks, comparable to or even outperforming human annotators (Gilardi et al., 2023). These prompting-based approaches offer a compelling alternative, eliminating the need for large training datasets required by fine-tuning methods (Wei et al., 2022). However, the potential of LLMs for legal NER, especially in Portuguese, remains underexplored.

Given the dual context of limited annotated legal corpora and the growing efficacy of LLMs, this work poses the following main research question (MRQ): *How to use Large Language Models in the Legal Named Entity Recognition task?*. Moreover, the process developed to resolve the MRQ has to be reliable and replicable, and the resulting annotations must be validated.

To address this, we propose a new annotation process for legal NER using LLMs with few-shot in-context learning. Our study takes as a reference the annotation performed in Correia et al. (2022) and the corpus presented by them. Additionally, we investigate the following research sub-questions: (RQ1) *How do we select a valuable set of examples for the prompt engineering process?* (RQ2) *What number of examples should be used to perform the task?* And, (RQ3) *How do we evaluate the annotations generated by the LLMs?*

To answer the *RQ1* and *RQ2*, we assessed the annotation capabilities of six open- and closed-source LLMs, at a coarser-grained level and in a strict- and relaxed-match way (Li et al., 2023a). The experiments were conducted in two phases. In the first, five documents (totaling 17,568 tokens, with 337 annotations) were annotated, also testing model sensitivity to in-context examples through three selection strategies: (i) random, (ii) similarity-based, and (iii) clustering-based. Furthermore, in the second phase, the best-performing configuration from the first phase was applied to 53 additional documents (134,711 tokens, 2,585 annotations).

We found no significant performance differences among the example selection strategies. Additionally, increasing the number of examples resulted in overall performance improvements for two models but degraded the performance of one. Moreover, we observed that smaller models struggled to recognize and annotate the entities correctly, while larger models surpassed an F1 score of 0.70 in the first phase, and the best model achieved 0.76 in the second phase. Thus, LLMs can generate good-quality annotations by recognizing most entities in the decisions. They may also be a valuable asset in helping annotators with the annotation process.

In summary, the main contributions of the paper are as follows:

- We present a new legal named entity recognition process that leverages LLMs' in-context few-shot learning capabilities.

- We present a reliable evaluation of the annotations generated, comparing them with those made by human annotators on the most extensive corpus known for legal NER.

- We assess the LLM's sensitivity to in-context examples by changing how they were selected and the impact of the number of examples included in the prompt.

This work is organized as follows: Section 2 describes the corpus used to evaluate the LLMs' annotation capabilities. Section 3 presents the related work that guided and founded this work. Section 4 details the proposed legal NER process using LLMs, including the examples database, prompt construction, and selection strategies. Section 5 presents the experiments and a detailed analysis of the results. Finally, Section 6 concludes with the main contributions and research questions addressed. Other supplementary information, such as details of the entities, prompt, models pricing, statistical tests, and experiments, can be found in Appendix A.

## 2 DATASET

The dataset used in this study, developed in Correia et al. (2022), comprises 594 decisions issued by the Brazilian Supreme Federal Court between 2009 and 2018. Each decision is annotated on two hierarchical levels of named entity: a coarser level, with four high-level entities, and a fine-grained level, with twenty-four nested entities. However, our work focuses only on the four coarser entities:

- **Academic Citations**: represents a direct citation of a book, article, or journal on the decision, often used to support arguments in a ruling. Such citations provide valuable insights into the influence of specific authors in the legal debate.

- **Legislative references**: citations to legislative references, a fundamental part of legal reasoning, and consist of mentions to articles, laws, sections, and constitution references.

- **Precedents**: citations to prior decisions present in the ruling, representing valuable information that proxies how relevant a given legal procedure was, is, and will be.

- **Persons**: this entity represents name, surname, titles, and treatment pronouns, as long as followed by the name, and not within other entities. The primary purpose is disambiguation regarding personal identification on the other entities.

The corpus contains 9,108 annotations of *precedents* with an average length of about 10.27 tokens, 1,775 *academic citations* averaging 24.60 tokens, 10,229 *legal references* averaging 8.41 tokens, and 11,943 *person* entities with an average of just over 3.38 tokens. For more information and examples of each coarser entity, see the Appendix A.1.

## 3 RELATED WORK

The emergence of LLMs has drawn significant attention from the scientific community due to their powerful in-context, few-shot learning capability. Recent works investigate the use of LLMs on the NER task and also propose co-annotation systems to leverage the strengths of both LLMs and humans. Aldeen et al. (2023) evaluates LLM data annotation capabilities across ten different datasets, using three distinct prompt strategies, one of which provides a task and labels description, as in our work. The results show that GPT-4 outperforms GPT-3.5 across various datasets, and the effectiveness of the prompt can vary depending on the task and the nature of the data.

An additional factor influencing model behavior is the temperature parameter, which regulates the level of randomness in generation and, therefore, affects annotation consistency. In Aldeen et al. (2023), no significant differences were observed in GPT-3.5 performance between using 0.25 or 1 temperature, as corroborated by Renze (2024), which investigates temperature sampling, varying from 0 to 1, over nine LLMs; and by Li et al. (2025) that tested open source models ranging from 1B to 80B parameters, for In-Context Learning and Instruction Following abilities. Based on these findings, our work adopts a fixed temperature of 0 for all experiments to provide a more constant output without compromising performance.

Regarding the use of LLMs on NER tasks, Xie et al. (2023); Aldeen et al. (2023) explores zero- and few-shot learning in a decomposed strategy that breaks the NER task into simpler subtasks, each handled in a separate dialogue iteration. Given the inherent complexity of legal texts, our work also adopted a decomposed strategy, but in a different approach in the recognition process, which is not a multi-turn dialogue. The annotations are created separately and then combined into a single document. Additionally, we also followed a few-shot strategy, based on the author's findings that it yields measurable improvements over the zero-shot baseline.

Furthermore, Wang et al. (2025) adapts LLMs to NER by transforming the sequence labeling task into a generation task, instructing models to generate labeled sequences by wrapping entities with special marks. In line with their work, our work adapts the legal NER tasks into a sequence labeling format. Through prompt engineering, we conduct the LLM to generate the same sequence, adding special symbols to mark the legal entities.

Conversely, works like Li et al. (2023b) and Wang et al. (2024) focused on creating a framework of collaboration between human annotators and LLMs, either by allocating some data for an LLM to annotate (Li et al., 2023b), or using human annotators to re-annotate a subset of LLM-generated labels (Wang et al., 2024). Similar to these works, our work proposes a process that leverages the LLMs' annotation capabilities to help annotators by reducing their workload.

## 4 METHODOLOGY

Our proposed methodology for annotating legal entities using LLM consists of two steps, as presented in Figure 1. We start by gathering high-quality examples through human intervention to build what we call the Minimal Golden Dataset (MGD). Afterward, we extract sentences containing at least one of the defined entities to compose the Example Database, which will be used to configure the few-shot. In the second step, from an input of documents to be annotated, we break each document into a set of excerpts. For each excerpt, we selected a set of examples using one of the three selection strategies (random, similarity-based, and clustering-based) to prompt construction, dealing with each entity separately (one prompt per entity). Finally, we compile all the responses into a single document, consolidating the annotations for all entities [1].

---

[1]In cases where annotations of different entities overlap on the same tokens, a heuristic resolution is applied based on a predefined priority hierarchy: Person, Legislative Reference, Precedent, and Academic Citation. Under this scheme, higher-priority entities prevail; for example, if a Person annotation overlaps with an Academic Citation, the latter is selected.

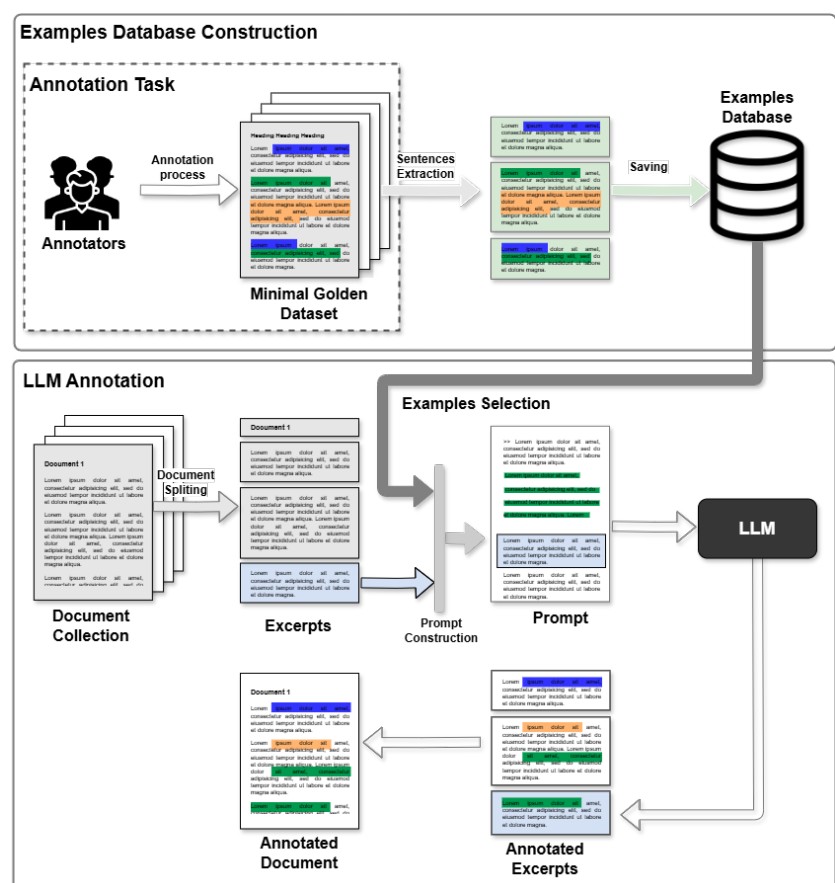

Figure 1: Overview of the proposed process for legal entity annotation with LLM.

## 4.1 MINIMAL GOLDEN DATASET (MGD)

The MGD is a small, high-quality annotated corpus specifically designed to serve as a basis for building the Example Database used in the few-shot learning process. It is composed of a limited number of manually annotated documents that collectively cover all target entity classes. Since the examples provided to the LLM are derived from the MGD, the dataset quality directly impacts the model's performance. Therefore, the involvement of domain specialists in this step is essential to ensure accuracy and reliability.

The size of the MGD, in terms of the number of documents and tokens, may vary according to the annotation task domain. In this study, we evaluated the impact of the number of examples on performance, and our results (described in subsection 5.3) showed that a few dozen examples are enough for our annotation task. Therefore, its limited size implies a reduced effort, time, and human resources in its construction.

## 4.2 EXAMPLES DATABASE

The Examples Database is constructed from the MGD by compiling the annotations by the annotators and grouping them by entity classes. At this point, several steps of cleaning and selection can be applied to ensure a high-quality set of examples. See the Appendix A.4 for the details.

In summary, each entity has its own examples set, and the examples are sentences without repetition that has at least one annotation. Thus, the Examples Database can be more formally defined as a collection of tuples: one element corresponds to the masked example, containing no information

about the annotation class or special marks, while the other represents the corresponding example with the annotation marks included.

## 4.3 PROMPT CONSTRUCTION

The developed prompt begins with contextual information about the task, followed by an explicit description of the entity, adapted from the guidelines in Correia et al. (2022), to enhance comprehension of the entity (see Appendix A.2). It also specifies the annotation procedure, requiring each entity to be marked with the beginning tag *"@@"* and the end tag *"##"* as in Wang et al. (2025).

Furthermore, we also include a few input-output examples demonstrating the task in the prompt. For illustration, for the excerpt *"As Súmulas 282 e 356 do STF dispõem respectivamente"*[2], which contains one Precedent citation, we have the following response: *"As @@Súmulas 282 e 356 do STF## dispõem respectivamente"*.

## 4.4 EXAMPLE SELECTION STRATEGIES

For LLMs' few-shot evaluation, we adopted three different strategies for example-selection: (i) **randomized**, where $k$ sentences are randomly chosen, each with equal probability of selection; (ii) **clustering**, the sentences are grouped into $k$ groups, matching the size of the few-shot set, and the centroid of each group is selected as an example (since the centroids had to be actual sentences, we used the *K-Medoids* model to create the clusters); and (iii) **similarity**, the sentences are encoded [3], and after that $k$ are selected based on cosine similarity with input.

## 5 EXPERIMENTS

This chapter describes experiments executed to evaluate LLMs in the proposed process and presents a detailed analysis of the results. Section 5.1 presents the data, models, metrics, and configurations; Section 5.2 reports the tuning experiments; Section 5.3 shows the best model's performance on a larger dataset; and Section 5.4 presents a manual review of its generated annotations.

## 5.1 EXPERIMENTAL SETUP

**MGD**. Based on the work of Correia et al. (2022), we obtained anonymized annotations from the two shorter training sessions of the annotators in the annotation activity and used them as MGD, resulting in a total of ten annotated documents. Although the documents used in the training sessions are also part of the final corpus, by relying on the preliminary annotations produced during the training session instead of the other documents from the corpus, we reduced the chances of contaminating the examples database, as the LLMs would only have access to annotations produced before the long final annotation effort, making our results more reliable.

**Example database**. From the MGD, we calculated the number of annotations for each annotator, excluding the annotators with none or very few annotations, and then the annotation's percentage of votes for the most voted class. After defining the annotation classes, we extracted the sentences containing at least one annotation for each entity, and filtered out extensive sentences to uniform the length for all four entities. A detailed version is presented in Appendix A.4.

**Language Models**. A total of six LLMs were selected, open- and closed-source. We tested the Gemini 1.5 Pro variant model and 1.5 Flash (Deepmind, 2024), GPT-4o mini (OpenAI, 2023), DeepSeek Chat V2 (DeepSeek-AI et al., 2024), Llama 3.1 405B and 70B (Dubey et al., 2024) (see the Appendix A.3 for pricing of each model). For all models, we set the temperature parameter to zero, which alters the level of randomness of the generation, providing a more constant output, for reproducibility purposes, and as suggested by Renze (2024), Aldeen et al. (2023) and Li et al. (2025), changes in temperature from 0.0 to 1.0 do not have a statistically significant impact on LLM's performances, so we discard changes in this parameter.

---

[2]The excerpt translation: *"STF Precedents 282 and 356 provide, respectively"*

[3]The sentences were embedded using a task-related encoder Liu et al. (2022), in this case, the Legal-BERTimbau (Souza et al., 2020) from Hugging Face, a BERT model fine-tuned with over 30,000 Portuguese legal documents available online. https://huggingface.co/rufimelo/Legal-BERTimbau-base

**Validation and Test Sets**. Due to the high cost of LLMs and the combinatorial number of examples, selection strategies, and models, we needed to reduce the number of documents used for evaluation. To do so, we performed a stratified selection of approximately 10% of the dataset, totaling 58 documents, to create the validation set comprising five documents and the test set, comprising 53 documents. Additionally, we split the decisions from both sets by sentence, with a minimum length of 2000 tokens. Table 1 presents the number of annotations and tokens for each entity on the validation set and for the test set.

**Metrics**. Since our purpose is to provide a reliable evaluation of LLM's performance in the legal named entity annotation task, we used the manual annotations on the validation and test set as ground truth. We evaluated performance with two criteria: (i) exact-match, where the LLM annotations boundaries and type must match the golden annotation; and (ii) relaxed-match, where the LLM annotations must assign the correct entity type and overlap with the ground truth regardless of boundaries Li et al. (2023a). Based on these criteria, we computed the precision, recall, and F1-score.

Table 1: Number of annotations and tokens for each named entity on the validation and test set

|  | Validation Set | | Test Set | |
| --- | --- | --- | --- | --- |
|  | Annotations | Tokens | Annotations | Tokens |
| Person | 144 | 505 | 916 | 3,137 |
| Legislative Ref. | 95 | 823 | 824 | 6,867 |
| Precedent | 80 | 1,013 | 693 | 7,410 |
| Academic Cit. | 18 | 532 | 152 | 3,897 |
|  | 337 | 2,873 | 2,585 | 21,311 |

## 5.2 Experimental Sweep on Validation Set

We conducted an exhaustive search on the validation set to find the best configuration for the number of examples and selection method. As stated before, we employed three selection methods: random, similarity, and clustering, and varied the number of examples to 4, 8, 16, and 32. This analysis was applied to each of the six selected models. Additionally, each combination was executed five times with different seeds, and the same examples were provided to all models. The Appendix A.6 presents an overview of LLMs' performances on this experiment.

As a result, we found that the Gemini 1.5 Pro achieved the best result among all LLMs, scoring over 0.80 F1, followed by its smaller version, Gemini 1.5 Flash, and Llama 3.1 405B, which yielded almost identical results for different selection strategies and numbers of examples, as well as DeepSeek V2. Lastly, Llama 3.1 70B presented a moderate performance, and GPT-4o mini struggled to annotate a significant portion of the entities and was the worst model tested.

As expected, some models perform better than others in following the task. Across all executions, Gemini Flash, Llama 405B, and Llama 70B produce more distant responses, especially for Academic Citation, even reaching a response only 60% similar. At the same time, the other three models are over 97%. This distance is associated with generation interruptions, where the model either fails to complete the sequences, includes unnecessary information, or hallucinates.

Moreover, the same behavior can be seen in the other part of the task, which involves correctly surrounding entities with special marks. Gemini Pro and Flash produced more than 600 marker errors with only four examples. Still, the number of errors dropped to fewer than 300 as the number of examples increased, with Gemini Pro achieving an 85% reduction using 32 examples. Llama 405B and 70B, as well as DeepSeek V2, presented the lowest rates of special marker errors. Notably, DeepSeek V2 generated zero errors with eight examples, and only one with 32. Finally, GPT-4o mini maintains an almost identical error rate across all executions.

The statistical test results (see Appendix A.5 for the details), applied to the F1-score, indicate that there are no significant differences between the selection methods. The order of examples, whether they are related to the input or not, does not affect the LLMs' performance. However, the number of examples produced varied effects: Gemini Pro and Flash improved their annotations with more examples, while no significant differences were observed in the Llama 405B, 70B, and GPT-4o mini.

Finally, DeepSeek V2's performance worsened with the increased context of using 32 examples compared to 16. The additional examples may act as noise, or the information vanishes as the context increases.

Based on the statements elicited before, the models can generalize even when using random examples, and their performance is independent of how those examples were selected. This scenario relies on the quality of the annotations that built the examples database; for a poorly set one, it may be necessary to tune the selection method and choose more carefully. However, the number of examples plays a more crucial role, as it affects the context length. Depending on the model, it may be sensitive to larger contexts, resulting in performance degradation.

Besides prompt configuration, the model's legal annotation capabilities remain associated with the data submitted during pre- and post-training steps, as highlighted by the statistical results of GPT-4o mini and Llama models. The performance of the three models was not affected by the selection method and the number of examples, suggesting that GPT-4o mini may have seen less legal data than the others, not enough to enable *judicial thinking*[4]. In contrast, Llama models demonstrated a priori *knowledge*: the larger one had a strong performance, achieved around 70% F1-score across all executions, even with fewer examples, while the 70B variant had a moderate one, likely limited by its smaller size (Brown et al. (2020)).

Finally, based on the significance testing and cost-benefit analysis of each model (see Appendix A.7 for details), the optimal selection strategy for all models is random selection. Moreover, for Gemini 1.5 Pro and Gemini 1.5 Flash, 16 examples are the outstanding number, as they provide similar performance to 32 examples but at a lower cost. For Llama 3.1 405B, Llama 3.1 70B, and GPT-4o Mini, four examples are sufficient, and it is way cheaper. Finally, DeepSeek performed better with 16 examples.

### 5.2.1 ENTITY ANALISYS

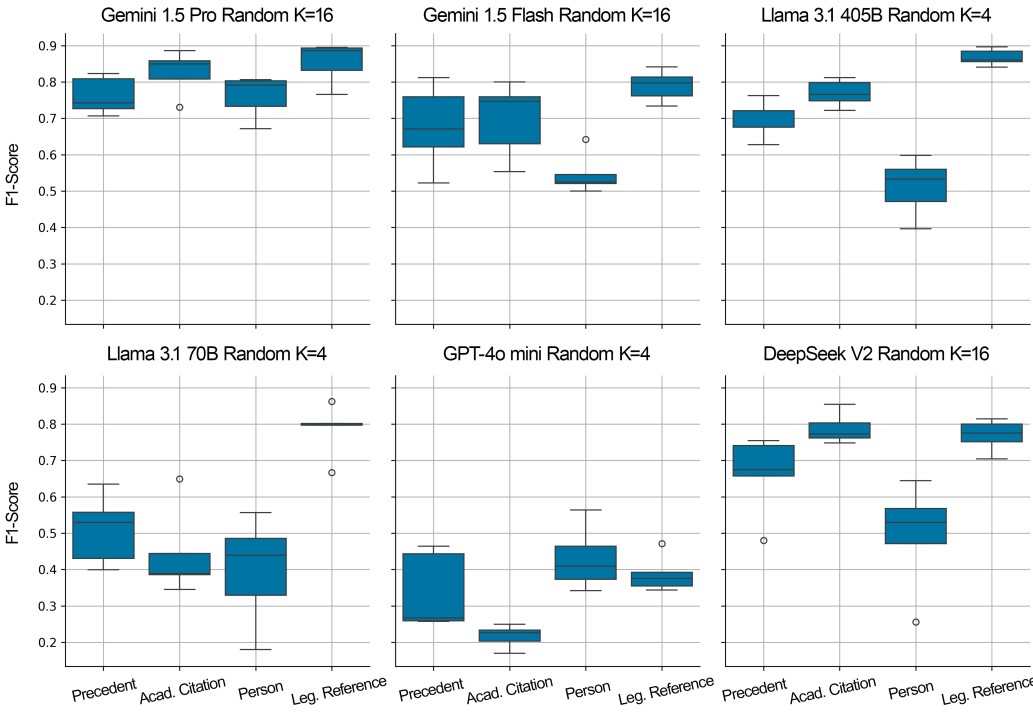

Figure 2: LLMs' average F1 performances with strict-match on the validation set for each entity using the best configuration of the number of examples and selection strategy.

---

[4]The term judicial thinking is not meant actual thinking, but the ability to derive to proper response by limiting the probability distribution based on in-context information.

Figure 2 displays the individual analysis for each entity and model, utilizing the optimal configuration. The Gemini 1.5 Pro once again achieved the best result, with a median F1-score over 0.70 for all entities, including the rarer ones, such as Academic Citation. The other models achieved results comparable to those of the Gemini 1.5 Pro in the individual analysis, except for the Person entity, which the other five models struggled with. For the entity Precedent, Legislative References, and Academic Citations, the Llama 3.1 405B, DeepSeek V2, and Gemini 1.5 Flash models exchanged positions for best performance. These results suggest that a multi-LLM strategy, assigning the model that performed best for each specific entity, could optimize both the accuracy and efficiency of annotation, reducing both cost and time in large-scale labeling tasks. See the Appendix A.8 for the results of mixing the LLM as independent agents by entity.

## 5.3 OPTIMAL MODEL ON TEST SET

To validate the experiment's results in the validation set, the Gemini 1.5 Pro was executed five times, with different seeds, on the test set using 16 examples and the random selection strategy. The results obtained with strict-match and for relaxed-match are shown in Table 2. With strict-match, the Gemini 1.5 Pro achieved an average F1-Score of 0.67, and with relaxed-match, 0.76, a significant improvement of 10% on average. Moreover, the annotation of Precedent entities was significantly worse. As shown in Table 2, the median of all the other entities is above 0.75, while the Precedent's median is close to 0.60. This result is related to the lack of a formal standard of precedent references in the Brazilian Supreme Court. However, the model performed well overall in the largest set, even with the increased variability of the entities.

Table 2: Gemini 1.5 Pro results on Test set with strict- and relaxed-match

|  | Strict-Match | | | Relaxed-Match | | |
| --- | --- | --- | --- | --- | --- | --- |
|  | Precision | Recall | F1-Score | Precision | Recall | F1-Score |
| Academic Citation | 0.69 | 0.71 | 0.69 | 0.72 | 0.81 | 0.76 |
| Leg. Reference | 0.71 | 0.69 | 0.70 | 0.77 | 0.74 | 0.76 |
| Person | 0.67 | 0.64 | 0.66 | 0.78 | 0.74 | 0.76 |
| Precedent | 0.62 | 0.56 | 0.59 | 0.64 | 0.58 | 0.61 |
|  | 0.68 | 0.66 | 0.67 | 0.74 | 0.74 | 0.76 |

## 5.4 MANUAL REVIEW OF LLM ANNOTATIONS

We conducted a manual review of the annotations generated by Gemini 1.5 Pro on the test set, focusing on cases where the model misclassified the same annotation across all five executions. The objective of this review was to assess the sources of error and limitations of the model, as well as to identify possible inconsistencies or errors in the manual annotation produced by Correia et al. (2022). To this end, 193 misclassifications cases were collected: (i) miss, ground-truth entities that the LLM failed to annotate, representing 46% of the cases; (ii) incorrect, LLM annotations where the labeled entity did not match the ground truth, with 30%; and (iii) spurious LLM annotations, corresponding to 22%.

The manual review task consisted of identifying who produced the correct annotation, the model, the annotators from Correia et al. (2022), or none of them; or as an issue stemming from the annotation guide (*e.g.*, in a particular case, the guide was ambiguous about how or which class to annotate it). The task was carried out by a team of five annotators, including one domain expert, and was divided into two sessions. In the first session, the annotators independently reviewed the same set of cases. Afterwards, in the second session, cases in which the annotators initially disagreed were reviewed until a complete consensus was reached regarding the correct label for each annotation.

The results show that the annotators of Correia et al. (2022) were correct in 69% of the cases. A common error found in LLMs' annotations was the misclassification of legislative references as precedents. However, although the annotators were correct in most cases, the LLM produced correct annotations in a notable 20% of instances, primarily involving entities that annotators failed to recognize. Finally, the manual review task also reveals cases where both annotations were incorrect, often due to entities not being completely covered. However, across all four entities, the Person

entity was the most common, demonstrating difficulty in annotating. The Appendix A.9 discusses examples of all three cases.

## 6    CONCLUSION

In this work, we aim to answer the **MRQ** by developing a process for legal entity annotation using LLM in-context few-shot learning capabilities, as presented in Section 4. To answer the **RQ1**, we developed three different strategies to select examples to be included in the prompt engineering process: a randomized selection, a similarity-based selection between input and examples, and a clustering selection to choose the most representative subset of examples. As a result, we found no significant difference between strategies implemented, as seen in Section 5.2.

We also evaluated the number of examples that should be used to perform the task, answering **RQ2**, and almost all six LLMs experimented achieved a remarkable performance with less than 32 examples, indicating that not many examples were necessary. In Section 5.2, we found that some models improved their performance by increasing the examples included in the prompt, such as Gemini 1.5 Pro and Gemini 1.5 Flash. On the other hand, the DeepSeek V2 worsened with increasing from 16 to 32 examples, suggesting that additional examples vanished or acted as noise, which oddly contradicts the experiments in DeepSeek-AI et al. (2024). Therefore, the impact of the number of examples depends on the models' architecture and pre-trained data.

Regarding the **RQ3**, we have compiled each entity annotation excerpt on a single document and compare them with the ground truth manually annotated by specialists in Correia et al. (2022), in two ways: in a strict-match manner when the type and boundaries of the generated annotation must match the ground-truth, and in a relaxed-match where only the types must match, despite the boundaries, as mentioned in Section 5.1. The first manner provides a more rigorous evaluation. However, since LLM annotations can be refined later, as proposed by Wang et al. (2024), or can serve only as highlights, the best way to evaluate annotations is using the relaxed-match.

Performing a few-shot NER in such a real-world problem is challenging and requires strong generalization abilities from the models. One of our conclusions is that the data on which the models were trained have a substantial impact, as it was observed that different models were better in different entities. Also, from our results, we observed that the larger models were better at recognizing and generating the labeled sequences than the smaller ones, noticing that lightweight models with less than 70B parameters could not annotate most of the entities.

Finally, our findings show that LLMs are indeed capable of making high-quality legal annotations in a few-shot learning context, even for those rarer entities as academic citations. Our best results are 0.76 F1 on average, scored by Gemini 1.5 Pro with only the entity description and a set of examples demonstrating the task. Thus, based on the observed results, LLMs can be a valuable tool in the annotation process by highlighting legal entities in the text, thereby reducing the burden, effort, and costs associated.

ACKNOWLEDGMENTS

[Omitted due to the double-blind review process.]

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

## A APPENDIX

### A.1 ENTITIES DESCRIPTION

#### A.1.1 ACADEMIC CITATIONS

This entity captures direct references to books, articles, and journals cited by justices to support their arguments. Although academic citations are rare regarding the other entities, they can offer valuable information on the influence of certain authors in the legal debate. An example of an academic citation is shown in Figure 3 [5] where the entity citation is annotated along with its nested elements, such as author *(MARINELA, Fernanda)*, title *(Direito Administrativo)*, edition *(6th)*, publication city *(Niterói)*, publisher *(Impetus)*, and year *(2012)*.

> *[...]. Trata-se de presunção relativa, do latim, presunção juris tantum, admitindo-se prova em contrário, cabendo o ônus probatório a quem aponta a ilegitimidade (...)." (MARINELA, Fernanda, Direito Administrativo, 6ª Edição, Niterói: Impetus, 2012).*

Figure 3: Academic Citation example with the author, title, and publisher information.

---

[5]The excerpt translation: "*[...]. This is a case of relative presumption, in Latin, presumption juris tantum, which admits evidence to the contrary, with the burden of proof falling on the party alleging illegitimacy (. . . )." (MARINELA, Fernanda, Administrative Law, 6th edition, Niterói: Impetus, 2012)*".

### A.1.2   LEGISLATIVE REFERENCES

These refer to legislative references, an essential component of legal reasoning, consisting of articles, laws, subsections, and constitutional references mentioned in the decisions. An example of legislative reference is shown in Figure 4[6] where the entity is annotated along with its nested elements, such as law *(Lei Nº 8.666/93)*, article *(71)*, and paragraph *(§1)*.

> *[...] com eficácia a partir da publicação da ata de julgamento  no Diário de Justiça eletrônico em 3/12/2010. Após essa data, qualquer decisão que negue a vigência a o §1º do art. 71 da Lei nº 8.666/93 por fundamento constitucional estará em confronto  com a decisão da Suprema Corte na ADC nº 16/DF*

Figure 4: Legislative Reference example with number, article, and the paragraph of the law.

### A.1.3   PERSONS

This entity represents name, surname, titles, and treatment pronouns as long as they are followed by the name and not within other entities. The primary purpose is disambiguation regarding the personal identification of the other entities. The Figure 5 [7] presents an example of a Person entity.

> *O SENHOR MINISTRO CELSO DE MELLO - (Relator): Trata-se de embargos de declaração opostos a decisão monocrática que, proferida em sede de recurso de agravo  (previsto e disciplinado na Lei nº 12.322/2010), dele não conheceu, em face de sua manifesta intempestividade.*

Figure 5: Person example.

### A.1.4   PRECEDENTS

This entity represents citations to prior court decisions referenced in the ruling. As cited in Leibon et al. (2018), precedent citation undoubtedly has great value in common-law-based judicial systems, such as in the United States and Canada, where courts are bound to their previous rulings. At the STF, such references do not follow a formal standard; they may appear with a legal procedure identification or temporal element like the judgment date or the decision's publication date to identify which decision is being referenced, or even both. These citations typically indicate that the referenced case has significant importance to the court Correia et al. (2022). An example of precedent is shown in Figure 6. [8].

---

[6]The excerpt translation: *""[...] with legal effect from the publication of the trial record in the Electronic Justice Gazette on December 3, 2010. After this date, any decision that denies the validity of §1 of Article 71 of Law No. 8,666/93 on constitutional grounds will be in conflict with the ruling of this Supreme Court in ADC No. 16/DF"*.

[7]The excerpt translation: *"THE HONORABLE MINISTER CELSO DE MELLO – (Reporting Justice): These are motions for clarification filed against a single-justice decision which, rendered in the context of an interlocutory appeal (as provided for and governed by Law No. 12,322/2010), was not admitted due to its clear untimeliness."*

[8]The excerpt translation: *"It requests the granting of a preliminary injunction to order the immediate suspension of the decision issued by the Fifth Panel of the Regional Labor Court of the Third Region, in the labor claim case No. 00573-2010-050-03-00-3. On the merits, it requests that the present claim be upheld, declaring the nullity of the challenged ruling."*

> *Requer a concessão de medida liminar para "determinar a suspensão imediata da decisão proferida pela Quinta Turma do Tribunal Regional do Trabalho da Terceira Região, nos autos da* reclamação trabalhista nº 00573-2010-050-03-00-3*". No mérito, requer seja julgada procedente a presente reclamação, declarando-se a nulidade do acórdão reclamado.*

Figure 6: Precedent example with the legal procedure class and number information.

## A.2 PROMPT ENGINEERING

The developed prompt described in Figure 7, first provides information about the task's context, followed by the few-shot examples, which will help the models to perform the annotation task, as system message and are represented by the red and yellow dashed lines. Finally, the input sequence for annotation is provided, as user message, represented by blue line. For the models that don't support system message, all the information is passed as a user message.

We are analyzing decisions published by Supremo Tribunal Federal
Academic Citations are text references to books, journals, and articles the judge writes on the ruling. An Academic Citation is composed of the following information:
Title - Identifies the paper title
First Author - Identifies the name of the first author
Co-Author - If it exists, identify the name of the other authors
Vehicle - Identifies the means through which the paper was published, such as journals, ISBN codes, etc.
Publisher - The name of the company responsible for literary works, discography, printed media publication, and distribution
Year of Publication - The year in which the paper was published
Your task is to annotate the academic citation entities contained in the text. The response must be a complete copy of the text with the addition of the annotation markers if it exist.

Some examples of the task are below:
Text: [...] a parte deve opor embargos declaratórios . Caso não o faça , não poderá invocar essa questão não apreciada na decisão recorrida ( RTJ 56/70 ; v . Súmula 356 do STF e Súmula 211 do STJ ; Nelson Luiz Pinto , Manual dos Recursos Cíveis , Malheiros Editores , 1999 , p . 234 ; Carlos Mário Velloso , Temas de Direito Público , p . 236 ) .
Response: [...] a parte deve opor embargos declaratórios . Caso não o faça , não poderá invocar essa questão não apreciada na decisão recorrida ( RTJ 56/70 ; v . Súmula 356 do STF e Súmula 211 do STJ ; @@Nelson Luiz Pinto , Manual dos Recursos Cíveis , Malheiros Editores , 1999 , p . 234## ; @@Carlos Mário Velloso , Temas de Direito Público , p . 236## ) .

Text: [...] interesse público sobre o interesse privado e o princípio da indisponibilidade do interesse público ( BANDEIRA DE MELLO , Celso Antônio . Curso de Direito Administrativo , 25ª Edição . São Paulo : Malheiros , 2008 . p . 55 ) , do que decorre o princípio constitucional da legalidade para a Administração Pública [...]
Response: LLM response

Figure 7: Prompt structure for Academic Citation entity annotation.

## A.3 LLMS PRICING

The Table 3 shows the input and output pricing for one million tokens for each model, on DeepSeek API[9] for DeepSeekV2, Gemini API[10] for Gemini models, OpenAI API[11] for GPT-4o mini, and DeepInfra API[12] for both Llama models.

---

[9]https://platform.deepseek.com/
[10]https://ai.google.dev/gemini-api
[11]https://platform.openai.com/
[12]https://deepinfra.com/

Table 3: LLMs pricing in US dollars per one million tokens. Accessed on July 25, 2024

| Model | Input Price | Output Price |
|---|---|---|
| DeepSeek V2 | $ 0,14 | $ 0,28 |
| Gemini 1.5 Pro | $ 3,50 | $ 10,50 |
| Gemini 1.5 Flash | $ 0,35 | $ 1,05 |
| GPT-4o mini | $ 0,50 | $ 1,50 |
| Llama 3.1 405B | $ 2,70 | $ 2,70 |
| Llama 3.1 70B | $ 0,52 | $ 0,75 |

## A.4 EXAMPLES DATABASE SETUP

From the MGD, we calculated the number of annotations for each annotator, excluding the annotators with none or very few annotations, and then the annotation's percentage of votes for the most voted class. Figure 8 presents the distribution of the annotations' percentage of votes for each entity after the annotator's removal.

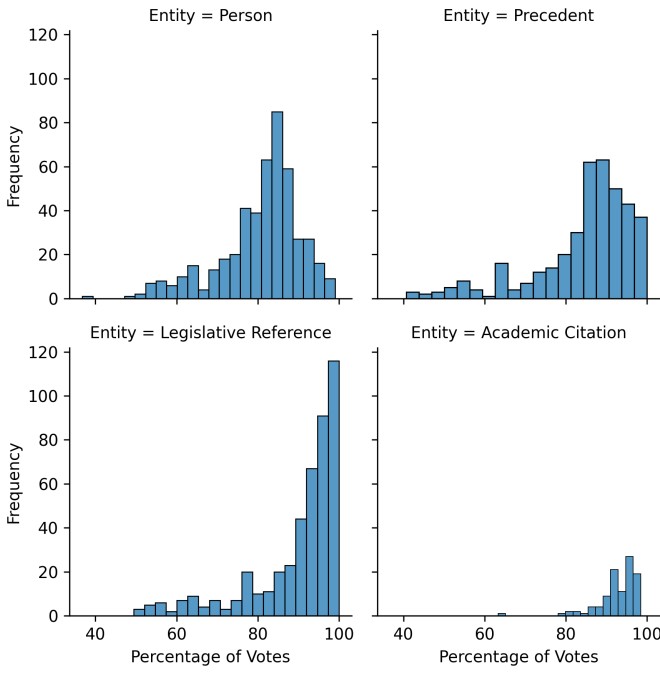

Figure 8: Percentage of votes for each annotation per entity.

After defining the annotations classes, we extracted the sentences containing at least one annotation for each entity. Figure 9a shows the distribution of the number of tokens in the sentences collected for each entity. The sentences containing academic citations tend to be more extensive regarding other entities. This is related to the length of academic citation, with more tokens on average, as present in Section 2. The median length of sentences for the other entities is below 100 tokens, but sometimes the number of tokens exceeds 700.

Although these more extensive sentences may appear commonly on a ruling, including them in the examples database will greatly increase the context sent to the LLMs, sometimes leading to an unpractical and unaffordable price. Thus, to address this problem and standardize the length of the sentences for all four entities, we apply a filter in the academic citations sentences by median, and the persons, precedents, and legislative references at the third percentile. The resulting distributions are present in the Figure 9b.

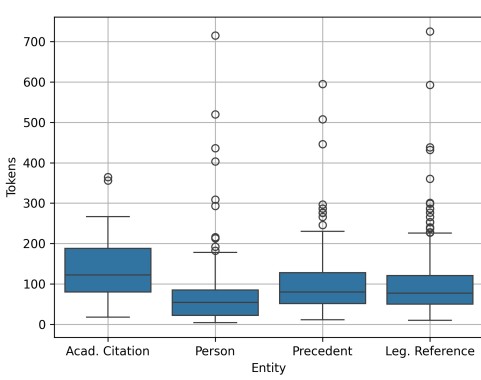 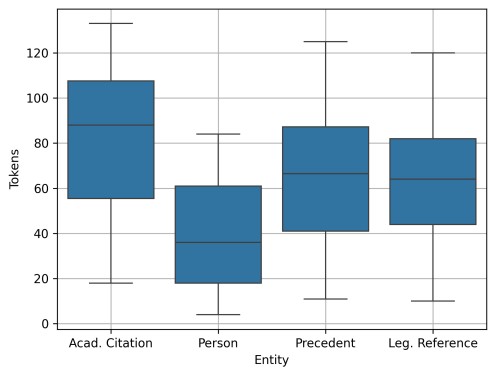

(a) Number of tokens in the sentence per entity before the filtering process.

(b) Number of tokens in the sentence per entity after the filtering process.

Figure 9: Number of tokens in the sentence for each entity.

Some observations regarding the annotations on MGD are as follows: some entities are easier to identify and distinguish from others. Most Academic Citation annotations receive over 80% of the annotator's vote — even so, there's no annotation with 100% of votes. The percentage of votes for the others is more spread. For the Person entity, some annotations received less than 40% of the votes, showing that most of the annotators did not recognize or distinguish it from the rest of the entities.

Moreover, the precedent annotations were mostly wrongly labeled as a person, followed by legislative references and academic citations. In legislative reference annotations, the precedent was the most common entity wrongly assigned, in second the person. The person appears again as the most wrongly assigned to the academic citation.

Therefore, even if the Person entity was created for disambiguation as presented in Section 2, this entity often led to misinterpretation and showed difficulty in annotating. Consequently, it will also be a challenge for the annotation process using LLMs.

## A.5 FINDINGS FROM STATISTICAL TESTS

To assess whether changes in the selection strategy and the number of examples led to statistically significant differences, we performed two variance statistical tests. Table 4 reports the p-values, degrees of freedom, and the test applied for each model regarding the example selection strategies, using a significance threshold of $\alpha = 0.01$. The results show that none of the models has a p-value lower than $\alpha$, rejecting the hypothesis that changes in selection strategies impact the F1-score. Table 5 and 6 presents the results regarding the variation in the number of examples. The Llama 405B, Llama 70B, and GPT-4 o mini models did not show significant differences, while Gemini 1.5 Pro and Flash, and Deepseek V2 showed performance shifts between 8 and 16 examples for the Gemini models and between 16 and 32 examples for DeepSeek V2.

| Model | Test | df | p-value |
|---|---|---|---|
| Gemini 1.5 Pro | Kruskal-Wallis | 2 | 0.682 |
| Gemini 1.5 Flash | Kruskal-Wallis | 2 | 0.632 |
| Llama 3.1 405B | ANOVA | (2, 57) | 0.023 |
| Llama 3.1 70B | ANOVA | (2, 57) | 0.133 |
| GPT-4o mini | ANOVA | (2, 57) | 0.416 |
| DeepSeek V2 | ANOVA | (2, 57) | 0.990 |

Table 4: F1-score statistical test for selection strategy on the validation set for $\alpha = 0.01$.

| Model | Test | df | p-value |
|---|---|---|---|
| Llama 3.1 405B | Kruskal-Wallis | 3 | 0.379 |
| Llama 3.1 70B | Kruskal-Wallis | 3 | 0.660 |
| GPT-4o mini | ANOVA | (3, 56) | 0.039 |

Table 5: F1-score statistical test for number of examples on the validation set for $\alpha = 0.01$.

| | Gemini 1.5 Pro | | | | Gemini 1.5 Flash | | | | DeepSeek V2 | | | |
|---|---|---|---|---|---|---|---|---|---|---|---|---|
| | Post-hoc Games-Howell | | | | Post-hoc Tukey | | | | Post-hoc Tukey | | | |
| | 4 | 8 | 16 | 32 | 4 | 8 | 16 | 32 | 4 | 8 | 16 | 32 |
| 4 | - | 0.286 | <.001 | <.001 | - | 0.229 | <.001 | <.001 | - | 0.404 | 0.140 | 0.687 |
| 8 | | - | <.001 | <.001 | | - | <.001 | <.001 | | - | 0.927 | 0.047 |
| 16 | | | - | 0.250 | | | - | 0.754 | | | - | <.001 |
| 32 | | | | - | | | | - | | | | - |

Table 6: F1-score statistical test for number of examples on the validation set for $\alpha = 0.01$.

## A.6 Overview of LLMs' performance on Sweep Experiment

The Figure 10 shows the average performances on the sweep experiment using strict-match.

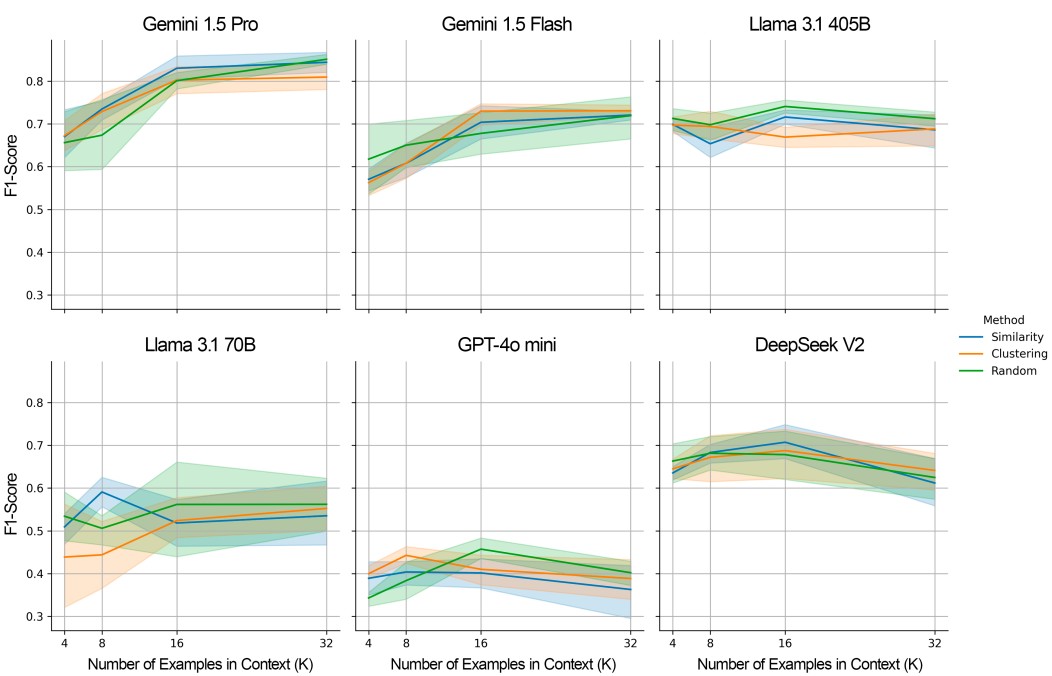

Figure 10: LLMs' average performances on the validation set with strict-match.

## A.7 Cost-benefit Analysis

We also calculate the cost-benefit ratio of each model by the number of examples in the context. To each $K$, we find the ratio between the response and the prompt, which includes the entity description, examples, and input text information. Finally, we calculate the costs for processing 1 million tokens for each LLM using the pricing presented in Section 5.1. As a result, alongside DeepSeek V2, the Gemini 1.5 Flash offers the best cost-benefit ratio with 16 examples, delivering quality annotations at a lower price, as depicted in Figure 11. Gemini 1.5 Pro was the most expensive model, regardless

of the number of examples. The larger version of Llma appears in the middle, with a strong performance and a moderate price. Finally, Llama 70B and GPT-4o mini cost as much as Gemini 1.5 Flash and DeepSeek V2, but have the worst performance.

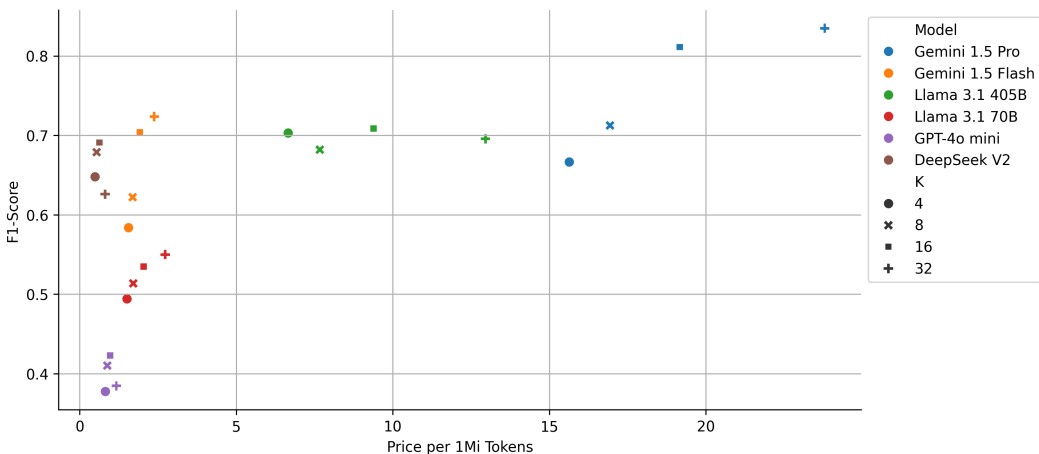

Figure 11: LLMs' Performance Cost-benefit on the validation set with strict match

## A.8 Multi-LLM Approach

Based on the results observed so far, we have also employed a cost-benefit multi-LLM approach for entity annotation. In this approach, we mix up the models Llama 405B for legislative reference annotation using four examples, DeepSeek V2 for academic citation using 16 examples, and Gemini 1.5 Flash for precedent annotation, following the individual analysis on Figure 2. For all models, the examples were selected using the random strategy.

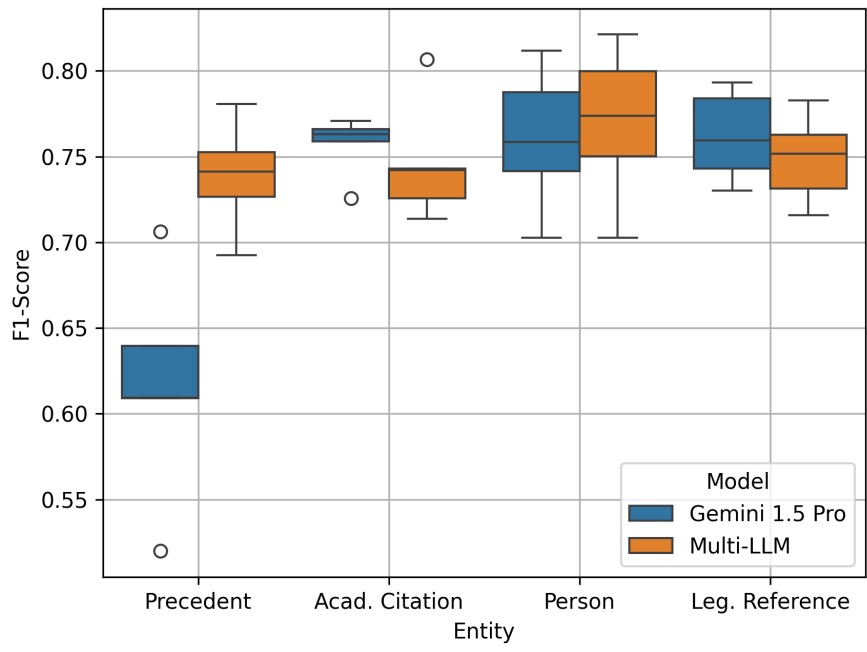

Figure 12: Multi-LLM approach performance on the test set with relaxed-match for each entity

Gemini 1.5 Flash's performance in Precedent annotation was close to Llama 405B, so we decided to keep it for diversity. Furthermore, as none of the three models performed well in annotating the person entity, we used the Gemini 1.5 Pro results.

The strict- and relaxed-match results are in Table 7. The approach using multiple LLMs yielded good results, with an F1 score of 0.74 using relaxed-match and 0.67 in strict-match. Furthermore, when analyzing the models individually, Llama 405B performed similarly to Gemini 1.5 Pro on Legislative Reference, with a median F1 score of 0.75, as shown in Figure 12, which is impressive considering only four examples were used. Additionally, DeepSeek V2 also demonstrated good performance on Academic Citation, similar to what was observed on the validation set, with an average F1 score of 0.74. Moreover, reaching an F1 score of 0.81. Gemini 1.5 Flash exceeded its validation set performance for Precedent and was also better than Gemini 1.5 Pro, showing a high capacity to generalize using the selected examples.

The differences observed in person entity annotation are due to the treatment of heuristic collisions: with fewer collisions occurring, more person annotations were preserved. These results highlight the importance of analyzing and utilizing different LLMs to build cost-effective solutions, targeting the best models for each entity. Additionally, it demonstrates that the process developed for annotating legal entities supports the addition of one or more LLMs during the annotation phase.

Table 7: Multi-LLM Approach results on Test set with strict- and relaxed-match

| | Strict-Match | | | Relaxed-Match | | |
|---|---|---|---|---|---|---|
| | Precision | Recall | F1-Score | Precision | Recall | F1-Score |
| Academic Citation | 0.64 | 0.72 | 0.67 | 0.66 | 0.83 | 0.74 |
| Leg. Reference | 0.76 | 0.60 | 0.69 | 0.79 | 0.69 | 0.75 |
| Person | 0.70 | 0.65 | 0.67 | 0.81 | 0.74 | 0.77 |
| Precedent | 0.78 | 0.66 | 0.66 | 0.82 | 0.68 | 0.74 |
| | 0.73 | 0.65 | 0.67 | 0.80 | 0.71 | 0.74 |

## A.9 MANUAL REVIEW OF LLM ANNOTATIONS

The manual review task took two weeks, and the results can be seen in Figure 13. The annotators of Correia et al. (2022) were correct in 69% of the cases, and most of these were entities not labeled by the LLM, and 37% were incorrectly assigned as other entities by the LLM. The primary mistake was in labeling the legislative reference as precedent. The sentence below is a fragment of an excerpt, recalling that decisions at the Jury Court regarding cases involving intentional crimes against life are made by majority vote, as specified in Article 488 of the Brazilian Code of Criminal Procedure. The confusion may be related to the a priori sense carried by the legislative reference, in addition to the number followed by an abbreviation form, frequently associated with precedent.

> Lembremos que as decisões do Conselho de Sentença são tomadas por maioria de votos *(art . 488 , CPP)*, mais um motivo para que não devesse devassar a votação , tomando público o placar (7 x 0 ; 6 x 1 ; 5 x 2 ou 4 x 3).

Another common mistake is regarding Complementary Laws, which often appear abbreviated, such as *LC 78/93*, and as mentioned before, this format matchs the precedent structure. Further, Complementary Laws were not explicitly mentioned in the legislative reference description, and their supplementary nature can hinder the model's correct activation.

Although the annotators were correct in most cases, the LLM produced correct annotations in a notable 20% of instances, primarily involving entities not included in the ground truth. The LLM was able to correctly annotate entities with extraction errors, such as *MP n˜ · 2. 200-2 /2001 de 24/0812001*, *AI˜232.439 - AgR / PB , Rel . Min . MAURÍCIO CORRÊA*, or *RTJ˜153/765*, all three with the escaped tild. Moreover, in some cases, the manual annotations of Correia et al. (2022) incorrectly labeled the Reporting Justice's name as a Person entity, while the model correctly followed the Precedent description in the prompt. For example, *Mandado de Injunção n. 284 , de redator para o acórdão Min . Celso de Mello*, the name Celso de Mello was annotated as a Person instead of a Precedent. In addition, the number of contracts, deals, and bid processes unrelated to

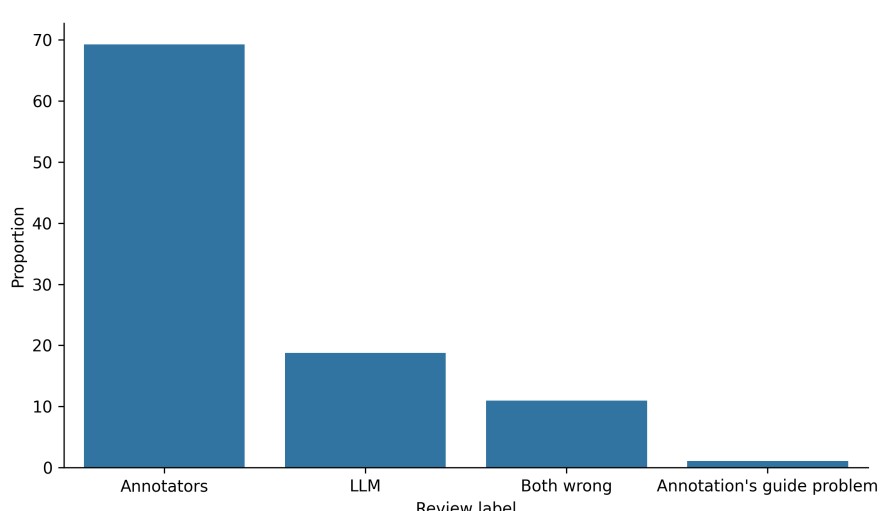

Figure 13: Proportion of each review label

the study case of Correia et al. (2022) was found on the ground-truth, misclassified as Legislative Reference or Precedent. The LLM ignored these appearances in the excerpt. These cases highlight the LLM's capabilities to support an annotation process, helping find entities along the excerpts and maintaining the correctness of the annotations.

The manual review task also reveals cases where both annotations were incorrect, which is often related to entities not completely covered, yet across all four entities, the Person entity was the most common, but it reveals difficulty annotating. In the following sentence, the annotators considered only the name Roberto Rosas without the title Professor, while the LLM annotation contains the opposite.

> " O ponto omisso, sobre o qual não foram opostos embargos de declaratórios, não pode ser objeto de recurso extraordinário, por faltar o requisito do prequestion-amento". A respeito do aludido verbete sumular, traz-se a lume comentário do **Professor Roberto Rosas**.

