# OpenReview forum: "Evaluating LLM In-Context Few-Shot Learning on Legal Entity Annotation Task"
_ICLR.cc/2026/Conference — Submitted to ICLR 2026_

### Official Review · Reviewer_9vtG · 2025-10-23

**Soundness:** 2
**Presentation:** 3
**Contribution:** 2
**Rating:** 2
**Confidence:** 4

**Summary:**

This paper evaluates the capability of Large Language Models (LLMs) for in-context few-shot learning on legal Named Entity Recognition (NER) tasks, specifically focusing on Portuguese legal documents. The authors propose a new annotation process that leverages LLMs to identify four types of legal entities: Academic Citations, Legislative References, Precedents, and Persons. The study uses the most extensive Portuguese corpus for legal NER and evaluates six different LLMs in various configurations. The experiments test different example selection strategies (random, similarity-based, and clustering-based) and varying numbers of examples (4, 8, 16, and 32) to determine optimal prompting approaches. The best-performing model achieved an F1 score of 0.76 using relaxed matching criteria. Additionally, a manual review of divergent annotations revealed that LLMs correctly identified entities missed by human annotators in 20% of cases, highlighting their potential to assist in the annotation process.

**Strengths:**

Practical application in a specialized domain: The research addresses a real-world challenge in legal text processing, particularly for non-English languages where annotated resources are limited.

Comprehensive evaluation methodology: The authors test multiple LLMs, example selection strategies, and example quantities, providing robust insights into optimal configurations for legal NER tasks.

Detailed error analysis: The manual review of annotation discrepancies offers valuable insights into both LLM limitations and potential improvements to existing human annotation processes.

Cost-benefit analysis: The paper includes a practical assessment of the cost-effectiveness of different models, considering both performance and computational expenses.

**Weaknesses:**

Limited language scope: While the focus on Portuguese legal documents addresses a gap in the literature, the findings might not generalize to other languages with different legal systems and terminologies.

Reliance on existing annotations: The evaluation uses human-annotated data as ground truth, but the manual review reveals inconsistencies in these annotations, which could affect performance metrics.

Lack of comparison with fine-tuned models: The paper doesn't compare the few-shot learning approach with traditional fine-tuned NER models, which would provide more context about the relative advantages of in-context learning.

Entity type imbalance: The dataset contains significantly fewer Academic Citations compared to other entity types, which could affect the reliability of performance metrics for this category.

Limited exploration of prompt variations: While the paper tests different example selection strategies and quantities, it doesn't explore variations in prompt structure or entity descriptions that might impact performance.

**Questions:**

How would the performance of the proposed approach compare to fine-tuned domain-specific NER models, and what are the trade-offs in terms of computational resources, data requirements, and accuracy?

How might the findings generalize to other legal systems and languages with different legal terminologies and citation formats?

Could the approach be extended to handle more complex nested entity structures, especially considering that the original corpus includes fine-grained nested entities?

How sensitive is the performance to variations in the entity descriptions provided in the prompts, and could optimizing these descriptions further improve results?

Given that LLMs correctly identified entities missed by human annotators in some cases, could an iterative annotation process that combines human and LLM inputs lead to higher quality annotated datasets?

---

### Official Review · Reviewer_4EvT · 2025-10-26

**Soundness:** 3
**Presentation:** 2
**Contribution:** 2
**Rating:** 4
**Confidence:** 4

**Summary:**

This work investigates how to leverage the in-context few-shot capabilities of LLMs for named entity recognition in the legal domain (legal NER) and proposes a complete annotation workflow: constructing a Minimal Golden Dataset (MGD) , building an Examples Database from the MGD , conducting few-shot prompting with three example selection strategies (random, similarity-based, and clustering-based) and different numbers of examples, constructing separate prompts for each entity and merging the results. The authors conducted comprehensive experiments on six open-source/closed-source LLMs (e.g., Gemini 1.5 Pro/Flash, Llama 3.1, etc.) using a Portuguese corpus based on decisions from the Brazilian Supreme Federal Court. Two evaluation criteria, strict-match and relaxed-match, were employed for performance comparison.

**Strengths:**

1. This work systematically applies the in-context few-shot method to Portuguese legal NER and validates it on a real-world, large-scale judicial corpus.
2. The paper conducts a systematic comparison of the sensitivity between "example selection strategies" (random, similarity-based, clustering-based) and the number of examples, and combines it with cost-benefit analysis, which constitutes a relatively practical contribution.
3. The experiments cover a wide scope: 6 different models, four entity categories, validation/test splits, repeated trials, and statistical tests (ANOVA, Kruskal-Wallis, post-hoc tests), featuring rigorous methodology and experimental design.
4. Manual review is conducted to analyze differences between model annotations and human annotations, providing qualitative insights rather than mere numerical comparisons.

**Weaknesses:**

1. Insufficient baseline evaluation: The paper fails to directly compare the performance of few-shot LLM with that of traditional supervised learning (fine-tuning on a small number of samples) or weak supervision methods.
2. Insufficient in-depth analysis of the "boundary" issue and marker error handling: There is a significant gap between strict and relaxed settings (with a lower strict score for Precedent), yet the paper mainly reports the overall F1 score and lacks fine-grained statistics on boundary error types (truncation, over-length, and misalignment). Additionally, only quantitative descriptions are provided regarding the sources of Marker (@@ … ##) errors and specific repair strategies.
3. Inadequate description of the dataset and generalization: Evaluation is only conducted on judicial documents from the Brazilian Supreme Federal Court (STF) (despite the large corpus size), while the model's robustness in other legal text genres, other jurisdictions, or under noisy conditions (e.g., OCR errors) is not assessed. The paper mentions that some typical formats (such as LC 78/93) may conflict with precedents, but no targeted data augmentation or pattern normalization methods are proposed.

**Questions:**

1. Complete Prompts and Examples: Could the complete prompts (including system/user instructions, the order of examples, and whether explanatory text is included) and sample outputs used for each entity be provided in the appendix or code repository? (This directly affects the reproducibility of both the experiment and prompt engineering.)
2. Reasons for GPT-4o mini’s Poor Performance: Could more specific diagnoses be provided (e.g., whether GPT-4o mini has truncation/API limitations, or if its training corpus is insufficient to cover Brazilian legal terminology)?
3. Bias of Heuristics for Boundary Handling: The paper uses a priority order (Person > Legislative > Precedent > Academic) to resolve overlaps. Could you demonstrate how this priority order affects the recall/precision of the final Person entity (e.g., does this heuristic lead to over-coverage or under-coverage of Person)? Additionally, what would the results be if this priority is replaced with other strategies such as "longest match first"?
4. Comparison of Manual Efficiency/Cost: You have provided the model’s token-based pricing and cost-benefit analysis. However, could you supplement this with a comparison of time/manual costs: after using LLM to assist with annotation, what is the average time required for manual correction? How does this compare to the total time/cost of pure manual annotation? This will directly support the claim of "cost savings."
5. Error Examples and Repair Strategies: During manual review, the model outperformed humans in 20% of cases (indicating that the model identified missed annotations or errors made by humans). Could these cases be categorized, and could automated repair suggestions be provided?

---

### Official Review · Reviewer_xozD · 2025-11-01

**Soundness:** 2
**Presentation:** 2
**Contribution:** 2
**Rating:** 2
**Confidence:** 4

**Summary:**

This paper studies the problem of legal entity recognition and presents an LLM-based method.

**Strengths:**

1. The topic is practically interesting and useful.
2. The authors have considered and tested several different LLMs.

**Weaknesses:**

1. This work is more engineering than research. The technical part is quite high-level. I don’t see research-level insights or designs (especially the rationale of designs), but only engineering-level descriptions and examples.
2. Limited technical depth regarding the standard of ICLR. More specifically, for RQ1, the authors investigated three simple strategies for prompt engineering with no significant difference. For RQ2, the authors simply tried 6 LLMs. Of course, one may try more, given sufficient time. For RQ3, the strict-match and relaxed-match methods are sort of standard routines in practice.

**Questions:**

See detailed comments.

---

### Official Review · Reviewer_i6zZ · 2025-11-01

**Soundness:** 2
**Presentation:** 3
**Contribution:** 2
**Rating:** 2
**Confidence:** 5

**Summary:**

Summary
The paper evaluates large language models (LLMs) for named entity recognition (NER) in Portuguese legal documents, focusing on four coarse-grained entity types. The stated motivation is to build a pipeline that could assist human annotators in labeling legal texts. The pipeline handles long-document segmentation and compares six off-the-shelf LLMs (both open and closed-source) under zero-shot and few-shot in-context learning. Results show that LLMs achieve competitive F1 scores even with random few-shot sampling, and that retrieval-based selection (RAG-style) offers no measurable benefit.

Overall Recommendation: Reject
While the paper is clearly written and applicable for this niche usecase at a surface level, it lacks originality, depth, and analytical rigor for ICLR. Its findings that LLMs perform reasonably on few-shot NER and that random sampling rivals similarity-based retrieval, are well-established in prior work. The human-in-the-loop (HiTL) framing is also not substantiated through workflow design, user studies, or quantitative cost analysis. The result is an incremental replication of known patterns in a narrow domain.

Reviewer LLM Usage:
I have read the paper in full and written the review myself. Large Language Models (LLMs) were used only for writing polish, clarity improvements, and to refresh memory of (public) related work or references. The analysis and conclusions are entirely my own.

**Strengths:**

1. Comprehensive model coverage: Evaluates six distinct LLMs, both closed- and open-source, providing useful comparative evidence.
2. Clear preprocessing pipeline: Long-document segmentation and sentence splitting are described clearly.
3. Interesting premise: The hypothesis that models trained on judicial corpora may encode “judicial reasoning” offers a speculative direction for future exploration.

**Weaknesses:**

1. Lack of novelty and analytical depth:
The paper confirms previously known findings: LLMs perform adequately for few-shot NER, and retrieval-based example selection rarely outperforms random sampling. No new prompting paradigm, retrieval method, or model adaptation technique is proposed.

2. Partial and shallow error analysis
Although the authors provide per-entity metrics, special-marker error counts, and a manual review of 193 misclassifications by five annotators, they stop short of deeper analyses such as confusion matrices, span-offset distributions, or nested-entity handling.

3. Unsupported HiTL framing
The work’s central claim, assisting human annotators, is not explored at all. There is no workflow description (pre-seeding, confidence based triaging to humans, uncertainty sampling, etc.), no usability study, and no productivity or cost metrics. The only cost discussion concerns API token pricing, not annotation effort. Thus, the HiTL style assisted annotations claim remains unsubstantiated.

4. Missing supervised baselines and limited data
No fine-tuned NER baselines (e.g., BERTimbau [Souza et al., 2020], XLM-R) are compared. BERTimbau is used only as an embedding encoder for retrieval. Moreover, evaluation covers just 5 validation documents  and 53 test documents, which is insufficient for stable generalization.

5. Narrow scope and simplified labeling
By restricting the task to four coarse entity types, the study avoids typical hierarchical NER challenges like nested entities, sub-type confusion, and overlapping spans. Consequently, the reported performance overestimates true real-world difficulty for their task which also has a second layer (which is considered out of scope).

6. Missing literature and context
The paper omits key prior work that already demonstrates effective few-shot and prompt-based NER: TANL [Paolini et al. 2021], PromptNER [Jie & Lu, 2022], InstructUIE [Wang et al., 2023] and cross-lingual benchmarks like XGLUE [Liang et al., 2020] and LEXTREME [Niklaus et al., 2023]. This omission overstates the novelty of the contribution.

7. Retrieval finding lacks explanation
Statistical tests (ANOVA and Kruskal-Wallis) confirm no significant difference between random, similarity, and cluster-based selection, yet the authors provide no analysis of retrieved-example quality or similarity distribution to qualitatively explain intuitively why this occurs.

8. Terminology
The phrase “mixing the LLM as independent agents by entity” simply denotes per-entity model ensembling not genuine multi-agent reasoning or tool use. So rigor in terminology is diluted.

**Questions:**

1. Replace references to “multiple agents” with “multiple LLM runs” or “ensemble of LLMs.”
2. Consider visualizing error types or entity overlaps for interpretability.
3. Also, consider exploring multiple LLM ensemble further.
4. Can you clarify whether few-shot examples were sampled from within the same document type or across different decision types, as this could impact generalization?

---

### Meta-Review · Area_Chair_JFv9 · 2026-01-07

**Summary:**

This paper explores the application of Large Language Models (LLMs) for Named Entity Recognition (NER) within the Portuguese legal domain, specifically focusing on judicial documents from the Brazilian Supreme Federal Court. The work evaluates six open and closed-source LLMs using in-context learning (zero-shot and few-shot). It investigates the impact of different example selection strategies—such as random, similarity-based (RAG), and clustering-based selection—across varying numbers of shots. The primary goal is to propose a pipeline that assists human annotators in a "Human-in-the-Loop" (HiTL) workflow to label coarse-grained legal entities like precedents, legislative references, and persons.

Almost all reviewers prefer a rejection primarily because the paper is perceived as an incremental engineering application rather than a contribution with sufficient research-level novelty or analytical depth for ICLR. The findings—such as the comparable performance of random versus retrieval-based example selection—are already well-established in existing literature, and the work lacks essential comparisons against traditional supervised baselines like fine-tuned BERT models. Furthermore, the central claim of a "Human-in-the-Loop" (HiTL) workflow remains unsubstantiated due to the absence of user studies, productivity metrics, or rigorous cost-benefit analyses regarding manual correction effort. This lack of depth extends to the evaluation itself, which suffers from a small, single-source dataset and a shallow error analysis that overlooks boundary misalignment and complex entity structures. Finally, the omission of key prior work on prompt-based NER and the use of imprecise terminology, such as mislabeling simple ensembles as "multi-agent" systems, further undermined the perceived rigor and originality of the submission.

The authors also didn't provide any response.

**Reviewer Concerns:**

see metareview

**Reviewer Scores:**

see metareview

---

### Decision · Program_Chairs · 2026-01-26

Reject